# Antimicrobial Resistant Pathogens Detected in Raw Pork and Poultry Meat in Retailing Outlets in Kenya

**DOI:** 10.3390/antibiotics12030613

**Published:** 2023-03-20

**Authors:** Patrick Muinde, John Maina, Kelvin Momanyi, Victor Yamo, John Mwaniki, John Kiiru

**Affiliations:** 1World Animal Protection, Westside Towers, 9th Floor, Suite 901, Along Lower Kabete Road, Westlands, P.O. Box 66580, Nairobi 00800, Kenya; 2Center for Microbiology Research, Kenya Medical Research Institute (KEMRI), P.O. Box 54840, Nairobi 00200, Kenya

**Keywords:** antibiotics, bacteria, isolates, antibiotic resistance, meat

## Abstract

There is increasing proof of bacterial resistance to antibiotics all over the world, and this puts the effectiveness of antimicrobials that have been essential in decreasing disease mortality and morbidity at stake. The WHO has labeled some classes of antimicrobials as vitally important to human health. Bacteria from animals are thought to be reservoirs of resistance genes that can be transferred to humans through the food chain. This study aimed to identify the resistance patterns of bacteria from pork and poultry meat samples purchased from leading retail outlets in Kenya. Of the 393 samples collected, 98.4% of pork and 96.6% of poultry were contaminated with high levels of bacteria. Among the 611 bacterial isolates recovered, 38.5% were multi-drug resistant. This resistance was noted for critically essential antimicrobials (according to the WHO) such as rifampicin (96%), ampicillin (35%), cefotaxime (9%), cefepime (6%), and ciprofloxacin (6%). Moreover, there was high resistance to key antimicrobials for veterinary medicine such as tetracycline (39%), sulfamethoxazole (33%), and trimethoprim (30%). It is essential to spread awareness about the judicious use of antibiotics and take preventive measures to reduce disease burden.

## 1. Introduction

Antimicrobial resistance (AMR) is a major global health concern, caused by the misuse and overuse of antimicrobials [1,2,3,4], which has led to microorganisms (including bacteria, fungi, viruses, and parasites) becoming resistant to the effects of these medications. The WHO defines AMR as the loss of susceptibility of these microorganisms to antimicrobials (such as antibiotics, antifungals, antivirals, and antiprotozoals). This imprudent use of antimicrobials in both the human and animal sector has resulted in the selection of pathogens resistant to multiple drugs.

It is now widely acknowledged that the rate of AMR development and spreading far outstrips the rate at which new antimicrobial drugs are being developed [5]. For instance, resistance to colistin, one of the last resort antibiotics used to treat multidrug-resistant Gram-negative infections, has been reported [6]. These multidrug-resistant (MDR) bacteria present a critical danger to public health. Such bacteria can survive the selective toxicity of antimicrobial use, enabling them to proliferate in clinical, on-farm, and environmental settings. For instance, patients infected with MDR bacteria tend to have a worse treatment outcome when compared to those infected with more susceptible organisms [7,8], in addition to being closely linked to the use of broad-spectrum antibiotics, both for empiric and definitive treatment [9].

AMR is a global issue that affects all nations including high, middle, and low-income countries, and it has increased the cost of health care and jeopardized gains made on goals set for Sustainable Development by 2030. A recent world bank report suggests AMR could cause low- and middle-income countries (LMICs) to lose more than 5% of their GDP and further push up to 28 million people, mostly in developing countries, into poverty by 2050 [10]. Globally, it is estimated that 1.27 million people die each year from drug resistant infections [11], which is projected to rise to 10 million deaths annually by 2050 [4]. By 2030, shocks due to antimicrobial resistance could cost the world up to USD 3.4 trillion a year [12], increasing to USD 100 trillion by 2050, with an overwhelming burden placed on LMICs [3].

Several antibiotic classes are used in both humans and animals, some of which are considered critically important to human health by the World Health Organization (WHO). Animal sectors chiefly use antimicrobials to prevent and treat infectious diseases [13] and promote growth in some countries [13]. Results from a recent analysis of global antibiotic sales data indicate that human antibiotic consumption is reported to have increased by 36% globally between 2000 and 2015, with most of the increase happening in LMICs where non-prescription use is still common [14].

In animals, antimicrobial use in animal production (especially in poultry and pigs) remains a key contributor to AMR [15]. The use is expected to increase exponentially due to the expansion of intensive production systems to meet the increasing demand for animal-sourced foods (ASFs), and the surge in disease burdens [16]. Over the next 20–40 years, the demand for ASFs will grow rapidly in Africa (meat consumption is forecast to grow by 30% by 2030) due to growth in the human population (from the current 1.2 billion to over 2.5 billion by 2050), increasing purchasing power and urbanization [17]. Across Africa, the current per capita annual consumption of meat and milk is about 14 kg and 30 L, respectively, and is projected to more than double to 26 kg and 64 L, respectively, by 2050 [18].

Of all antibiotics currently used in the world, approximately 73% are used within livestock [17], and a significant part is used for disease prevention (prophylaxis). The consumption is predicted to grow significantly by 2030 with the highest growth rates predicted within LMICs [19] because there will likely be a shift to more industrial livestock systems. The use of antimicrobials for growth promotion and routine disease prevention in groups of animals, without addressing the underlying animal welfare and husbandry practices that can prevent disease occurrences at farm levels, is contributing to the development and spread of AMR [20]. For instance, stressors have been documented to cause proliferation and colonization of pathogens such as *E. coli* O157:H7, *Campylobacter* and *Salmonella enterica*, which lead to fecal shedding as reported in pigs [21,22]. A review by Rostagno (2009) [23] reported how stress indirectly encourages the proliferation of enteric pathogens by suppressing the immune system and by physiological changes in the gastrointestinal tract via the action of the stress hormones.

Just like other countries, Kenya is already experiencing increasing levels of antimicrobial use and antimicrobial resistance. In 2013, it was estimated that 395 tons of antibiotics were used for food animal production in Kenya, of which 43% of them are classified as critically relevant by the WHO [16]. Notably, key antimicrobial-resistant foodborne pathogens such as *E. coli*, *Salmonella* and *Campylobacter* spp. have been documented in Kenya, with increasing frequency as causes of foodborne diseases of global public health significance [5,24]. Antimicrobial-resistant bacteria that originate in the gastrointestinal tract of animals can contaminate meat during animal slaughter and food processing or contaminate the environment with animal feces, and thus be transferred to humans through handling or consuming contaminated food or coming into contact with animal waste. This can lead to antimicrobial-resistant intestinal infections.

Studies have supported the hypothesis of the link between antimicrobial use in agricultural production systems and the emergence of AMR, especially the pronounced lack of biosecurity measures and low animal welfare practices. Since 1986, when the use of antimicrobials as growth promoters was banned in Sweden, the country has seen 65% considerable decrease in the utilization of antimicrobials in food animal production [25], resulting in a substantial reduction in the emergence and spread of AMR.

The demand for animal-based products has caused the intensification of production systems. As part of this intensification, antibiotics have been overused, both as growth promoters and as preventive measures. This has caused the emergence of pathogens that are resistant to antibiotics.

Antimicrobial-resistant bacteria have been reported in veterinary and food-related settings [26,27]. However, very few studies have assessed AMR in Kenya’s pig and poultry meat products. Therefore, we conducted this study to determine the presence of foodborne bacteria in pork and chicken products and the resistance profiles of the isolates to selected clinically relevant antimicrobials.

## 2. Results

### 2.1. Sample Distribution across the Retailing Outlets

Table 1 shows that out of the 393 pork and poultry samples, 107 (27.2%) were obtained from an international outlet, followed by one regional outlet. The majority of the samples (53.4%: *n* = 210) were taken from the fridge/freezer and fresh meat section (44.8%: *n* = 176), while only seven (1.8%) samples were acquired from the supermarket shelves. Nairobi accounted for most of the samples collected (nearly 75% of the pork and 63% of the poultry). This is because most of the supermarkets are concentrated in Nairobi, with a few in other big towns in Kenya. For example, at the time of sampling, the publicly available information in their websites showed that 89% of international supermarket outlets were in Nairobi, while 37% of the regional supermarket outlets were found in Nairobi, with the rest situated in other big towns which include Mombasa, Kisumu, and Nakuru.

The study anonymized the identities of the six retail outlets by assigning each an alphabetic designation from A to F (Table 1). Additionally, the outlets were classified into local, regional, or internationally based on criteria such as franchise status, ownership, and geographic reach.

### 2.2. Prevalence of Isolated Bacterial Contaminants

Nearly 98.4% (184/187) of pork and 96.6% (199/206) of poultry samples tested revealed the presence of at least one type of bacterium. In total, 611 bacterial isolates were recovered from the analysis of the 393 pork and poultry samples, but only 551 isolates were processed further depended on the resource availability. The majority (50.9%) of the isolates were detected in poultry samples but the difference was not found to be statistically significant (*p* = 0.157). Escherichia coli was the most common Gram-negative bacteria in both pork and poultry samples, at 47.7% and 49.2%, respectively. Meanwhile, *Staphylococcus* spp. (Gram-positive) was found in 28 (9.3%) of pork samples and 13 (4.2%) of poultry samples. However, it was difficult to identify *Staphylococcus aureus*, which is known to cause staphylococcal food poisoning, as the classical biochemical methods used were limited to identifying the genus level. Additional isolates included *Klebsiella* spp. (19.1%), *Salmonella* spp. (17.8%), *Shigella* (7.5%), and *Pseudomonas* spp. (0.3%), as seen in Figure 1.

### 2.3. The Overall Antimicrobial Resistance Profiles

Out of the 611 total isolates, 551 were chosen for analysis of antimicrobial resistance based on the sample type, retail store, and resources availability. The results, shown in Figure 2, revealed that rifampicin had the highest resistance rate of 96%. Ampicillin, sulfamethoxazole, trimethoprim, and tetracycline had resistance rates of 30–39%. The least resistance was seen with gentamicin (3%), cefepime (6%), ceftazidime (6%), and ciprofloxacin (7%).

As illustrated in Figure 3 below, isolates from chicken and pork samples exhibited the same resistance rate of 96% toward rifampicin. However, chicken isolates demonstrated higher resistance than those from pork against tetracycline (47% vs. 31%), sulfamethoxazole (41% vs. 26%), and trimethoprim (37% vs. 23%). Conversely, the resistance rates of pork isolate to ampicillin (35% vs. 34%), amoxicillin-clavulanic acid (19% vs. 12%), and cefoxitin (26% vs. 20%) were higher compared to that of chicken isolates.

### 2.4. The Antimicrobial Resistance Profiles of the Recovered Isolates

The comparative analysis of antimicrobial resistance (AMR) profiles was conducted using two *Pseudomonas aeruginosa* isolates, which had 100% resistance to eight antibiotics tested. Among the Gram-negatives, *Klebsiella* spp. had the highest level of resistance, particularly toward tetracycline (46%), sulfamethoxazole (43%), trimethoprim (37%), and cefoxitin (15%). Furthermore, this isolate also exhibited moderate resistance toward expanded spectrum antibiotics such as cefotaxime (5%), ceftazidime (3%), and cefepime (3%). As for *Staphylococcus* spp., its resistance profiles were higher than all other isolates for all antibiotics except rifampicin, which was higher in *Klebsiella* spp. (99%). *Salmonella* spp. was the least resistant, ranging from 1% to 31%, with the exception of rifampicin, which was at 97%. Table 2 shows the antimicrobial resistance profiles of the isolates mentioned above, *E. coli* and *Shigella* spp.

Out of the 551 isolates tested, 32.1% (177) were fully susceptible to the 14 antimicrobial agents in the 6 classes. *Shigella* had the highest number of fully susceptible isolates at 41.7% (15/36). The prevalence of multidrug resistance (MDR) was 16.2%, 6.9% of the isolates were resistant to 3 classes, 4.5% to 4 classes, 3.8% to 5 classes, and 0.9% to all 6 classes. In addition, 100% of *P. aeruginosa* (2/2) isolates and 76.1% of *Staphylococcus* spp. (32/42) isolates had MDR, while 15.8% of *Klebsiella* spp. (15/95) isolates had MDR (Table 3).

## 3. Discussion

The results of this investigation give an excellent glimpse of the levels of bacterial carriage in chicken and poultry meat sold at major supermarkets across Kenya. This study noted a high prevalence of bacteria often considered commensals [29], such as *E. coli* (48.4%) and *Klebsiella* spp. (19.1%)*,* and foodborne pathogens, such as *Salmonella* spp. (17.8%) and *Staphylococcus* spp. (6.7%). In addition, the study isolated *Shigella* spp. (7.5%) and *P. aeruginosa* (0.3%), bacteria associated with severe gastrointestinal infections such as chronic diarrhea and enterocolitis [30]; [31]. A similar study by Wardhana et al. (2021) [32] also reported a high prevalence of *S. aureus* (58.3%), *Salmonella* spp. (48.3%), and *E. coli* (40%) in retail chicken samples in Indonesia. Furthermore, a prevalence of 58.1% in *Salmonella* spp. [33], 18% in *E. coli* [34], 11.5% in *P. aeruginosa* [35], and 5.6% in *S. aureus* spp. [36] has been reported in pork sample from retail markets.

Though there was a potential of cross contamination in the fridge/freezer shelves through liquid drips from one food item to another, the likelihood of this happening was reduced because the samples were found to be shrink wrapped in polymer plastic film bags at the time of sampling. Therefore, the reported bacterial contamination of pork and chicken meat might have its origins at the farm level during the slaughtering process or packaging.

The extensive use of antibiotics for prevention and growth promotion in chickens and pigs has been a major factor in the development of antimicrobial resistance in bacteria with zoonotic potential, which is a serious public health issue [37]. For instance, according to a recent study by Ndukui et al. (2021), oxytetracycline (85%) and Amoxil (88%) are widely used antibiotics in commercial chicken raising in Kenya [38]. Our study findings of 39% and 35% frequencies against tetracycline and ampicillin are possibly a reflection of the implications of heavy antibiotics usage, as reported previously [38].

The development of antibiotic resistance to broad-spectrum medications such as ciprofloxacin, gentamicin, and cefepime, which are all on the WHO list of critically important antimicrobials for human medicine, is a growing concern. These medications provide limited alternatives, and it may be difficult to treat bacterial infections that do not respond to them with readily available drugs. The situation is further exacerbated by the increasing resistance to amoxiclav, ceftazidime, and gentamicin, which has risen to 16%, 6% and 3%, respectively, from levels of 2.6%, 0% and 0.6% reported in a similar study in Kenya less than 10 years ago [39]. The resistance to sulfamethoxazole and trimethoprim, two highly important antimicrobials used to treat bacterial and coccidial infections in humans and animals, has risen alarmingly. Furthermore, the resistance to tetracycline, ampicillin, amoxicillin-clavulanic, and ciprofloxacin, which are widely used to treat septicemia and respiratory infections in livestock, has reached 39%, 35%, 16% and 6%, respectively. These results could suggest that the bacteria in poultry and chicken farming sectors have developed resistance to antimicrobials due to heavy usage. Even though the use of rifampicin is prohibited or limited in many countries, it still remained the most resistant in all the isolates, which may indicate its use for prophylaxis purposes in the livestock sector. The reported high resistance towards rifampicin is expected considering the antibiotic is not recommended and is conventionally less active against infections caused by Gram-negative bacteria. Nonetheless, the resistance to antibiotics in chicken and pork isolates was found to be similarly high, which emphasizes the need for stewardship in chicken and pig farming and proper hygienic handling to avoid microbial contamination.

Though strains of *E. coli*, which was the most isolated bacteria in this study, are not harmful, some strains have acquired traits such as toxin production, making them pathogenic [40] and capable to cause serious foodborne pathogens. *Klebsiella* spp. and other known pathogens such as *Salmonella* and *Shigella* spp. were also found to be highly resistant to antibiotics commonly used to treat foodborne illnesses, for example ciprofloxacin (5% and 6%, respectively). Notably, *Pseudomonas aeruginosa*, which the WHO has identified as a critical pathogen due to its high resistance to antimicrobials, was among the most resistant to the antibiotics tested in chicken and pork samples.

In a similar study conducted in Kenya [41], in Vietnam, Salmonella isolates from chicken and pork samples exhibited a lower level of resistance to ampicillin (15%), tetracycline (36.7%), nalidixic acid (12.0%), and chloramphenicol (10%) than the corresponding rates seen in *E. coli* (85%, 66.7%, 24.1%, and 14.8%, respectively). Additionally, the resistance levels of ceftazidime, cefepime, ciprofloxacin, and tetracycline were reported to be 4.4%, 0.9%, 21%, and 66.4%, respectively [42]. It is alarming that 16.2% of the bacteria isolates studied were resistant to different antibiotics, as it jeopardizes the efficacy of antibiotic treatment for foodborne illnesses.

## 4. Materials and Methods

### 4.1. Sample Collection

For this cross-sectional study, we collected a total of uncooked 187 pork and 206 chicken samples between April and July 2020 from six leading supermarkets across five towns (Nairobi, Kisumu, Nakuru, Nanyuki and Eldoret) in Kenya. The leading supermarkets in Kenya are concentrated in cities (Nairobi, Kisumu, Mombasa, and Nakuru) and other big towns such as Eldoret and Naivasha. This is because urbanization has created easily accessible market for them in addition to improved infrastructure that facilitates transportation and storing of the perishable animal-sourced foods in their outlets. At the time of conducting this study, publicly available information showed that 89% of the international supermarket outlets and over a third of regional supermarket outlets were in Nairobi, the capital city of Kenya, with a population of over 4 million people as reported in 2019 [43]. All the samples were purchased either as wrapped/sealed by the supplier or repackaged by the outlet, within their expiry date, and the product branding was covered to blind the laboratory personnel. The samples were then transported in coolers to the Kenya Medical Research Institute within five hours, where processing began immediately. Moreover, data on the freshness of the sample, packaging method, storage temperature, and the type of PPE worn by the supermarket attendant was also collected.

### 4.2. Processing of the Samples in the Laboratory

Laboratory tests were carried out to detect foodborne bacteria in poultry meat and pork samples. Enrichment strategies and media were chosen carefully to enable the growth of non-fastidious bacteria.

To do this, 10 g of the meat sample was added into 90 mL of buffered peptone water (BPW) contained in a sterile stomacher bag and then homogenized with a stomacher machine (Stomacher^®^ 400 Circulator). The resulting homogenate was transferred into as sterile 250 mL culture media bottle, loosely capped to allow growth of facultative bacteria, and incubated for 24 h at 37 °C. To grow Gram-negative (such as *E. coli*, *Shigella* and *Klebsiella* spp.) and *Staphylococcus* spp., a loopful (10 µL) of the BPW enrichment was streaked on MacConkey and Mannitol salt agar, respectively, and incubated overnight at 37 °C. Concurrently, 1 mL of the BPW pre-enrichment was added into 9 mL of Rappaport Vassiliadis (RV) enrichment media and incubated for 24 h at 37 °C to enhance enrichment of *Salmonella* spp. A loopful (10 µL) of RV enrichment was streaked on Xylose lysine deoxycholate (XLD), and incubated for 24 h at 37 °C, for isolation of *Salmonella* spp. Gram stain test was used to identify Gram-positive and negative isolates. Classical biochemistry methods were utilized to determine the species of the isolates. To identify Gram-negative bacteria, a pure colony was inoculated in triple sugar iron (TSI), lysin indole motility (LIM), methyl red voges Proskauer (MRVP), urea and citrate media and incubated overnight at 37 °C. To identify *Staphylococcus* species, we used the catalase test.

Antimicrobial susceptibility was then tested by the disc diffusion method with a selection of antimicrobials in line with the World Health Organization’s recommendations for each species. In addition to the recommended antimicrobials, we also added nalidixic acid, chloramphenicol, and rifampicin to the antibiotics list because they are rarely tested drugs against enteric bacteria. A fresh, pure overnight culture was used to make a 0.5-equivalent MacFarland standard suspension in sterile normal saline. The suspension was evenly spread on Mueller–Hinton agar plates and antimicrobial discs (Oxoid) dispensed on the surface, after which the plates were incubated at 35 °C for 16–18 h. The *Escherichia coli* ATCC^®^ 25,922 and *Staphylococcus aureus* ATCC^®^ 25,923 were used for quality control. To analyze the antimicrobial resistance profiles and multidrug resistance, WHONET 2022 software (https://whonet.org/software.html, accessed on 2 November 2022) using the CLSI breakpoints interpretation guidelines were utilized.

### 4.3. Statistical Data Analysis

Data were gathered quantitatively and qualitatively using Epicollect5 mobile and web applications and were then exported to SPSS Statistics Software^®^ (IBM Corp., Armonk, NY, USA, v.22) for statistical analysis. The chi-square test or Fisher’s exact test was used to assess differences, and a *p*-value of less than 0.05 was considered as significant.

## 5. Conclusions

To sum up, this research suggests a high risk of food safety concerns, with chicken meat and pork from both local and international supermarkets in Kenya being found contaminated with bacterial contaminants, potentially spreading foodborne illnesses. It is essential to enforce high standards of food hygiene and sanitation throughout the supply chain, especially at the time of slaughter and packaging, in order to prevent the introduction of bacteria to the food and the subsequent spread of foodborne pathogens. Although we did not establish the source of microbial contamination, it is essential for retailing outlets to adhere to hygienic principles when handling and processing pork and chicken meat products to reduce the potential risk of microbial contamination. This study revealed that the resistance to essential classes of antibiotics, such as cephalosporins, aminoglycosides, and fluoroquinolones, is not high; however, an analysis of similar data showed that resistance might be increasing over time. Moreover, the analysis of our data showed worrying levels of resistance to tetracycline and penicillin, two of the most commonly used antibiotics in animal agriculture, demonstrating the necessity of responsible antibiotic use, improved and humane animal production methods, and increased biosecurity levels. It is particularly concerning that a few isolates were resistant to more than three types of antibiotics, which could make it more difficult to treat foodborne illnesses and other diseases. With new resistance mechanisms emerging and spreading globally, there is a need for a concerted effort to gain insights on how to better tackle AMR as well as raise awareness.

## Figures and Tables

**Figure 1 antibiotics-12-00613-f001:**
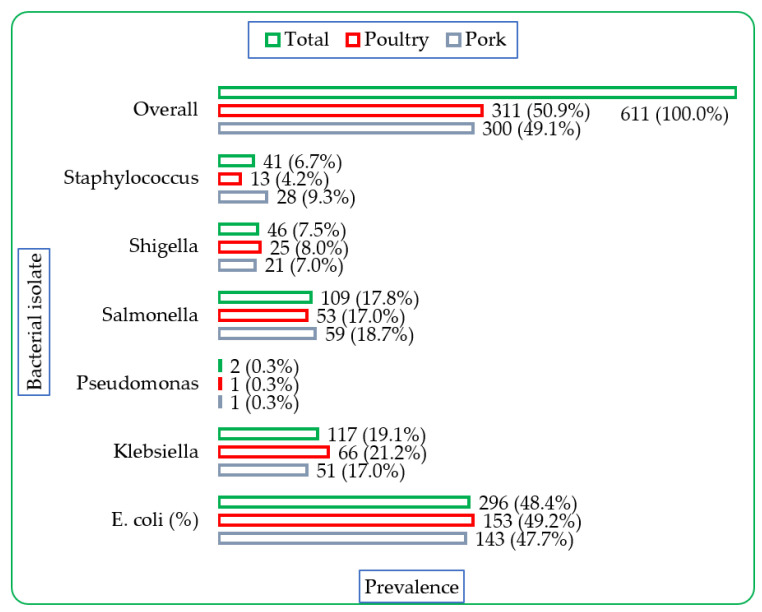
Figure showing the prevalence of bacterial contaminants that were isolated. The overall proportions for poultry and pork isolates were determined by analyzing the 611 total isolates recovered, which included 311 poultry and 300 pork.

**Figure 2 antibiotics-12-00613-f002:**
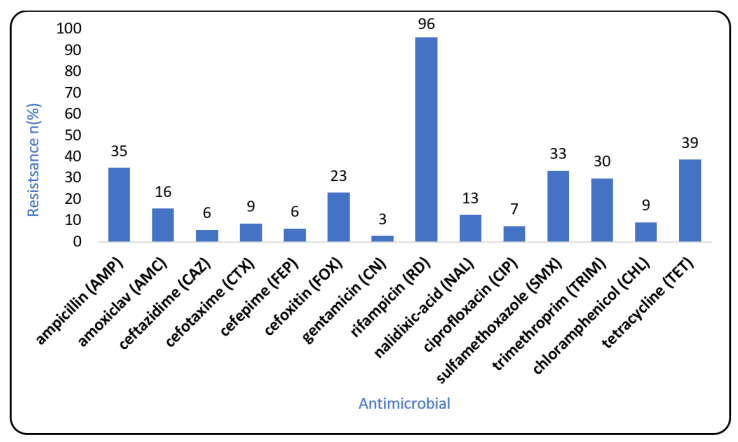
Figure showing the overall antimicrobial resistance profiles. The profiles of resistance were determined by examining 551 bacteria isolates taken from chicken and pork samples, comprising both Gram-positive and Gram-negative organisms. The percentage resistance was calculated by dividing the number of resistant isolates with the total number of test isolates.

**Figure 3 antibiotics-12-00613-f003:**
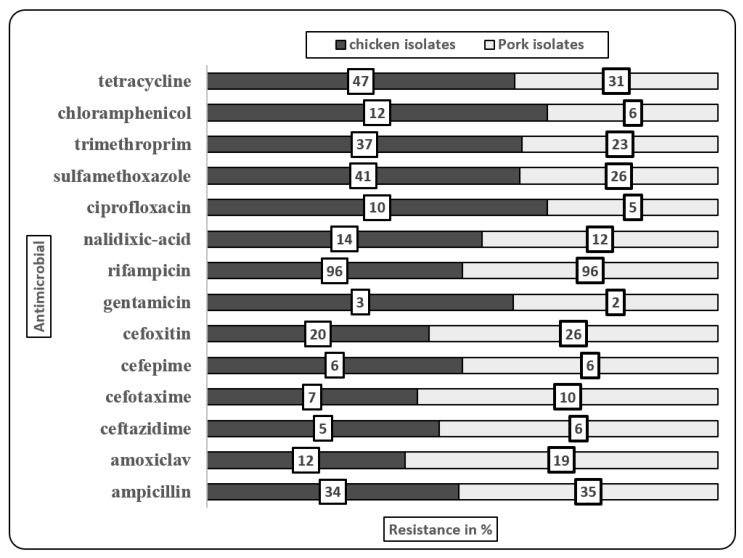
Figure showing the antimicrobial resistance profiles of chicken and pork isolates. The antimicrobial sensitivity testing is hinged on 268 chicken and 283 pork isolates.

**Table 1 antibiotics-12-00613-t001:** Table showing the sample distribution across the selected retailing outlets in Kenya.

Type of Supermarket	Supermarket Name	No. of Poultry Samples Collected	No. of Pork Samples Collected	Total
International	A	31 (15.0%)	76 (40.6%)	107 (27.2%)
Regional	B	64 (31.1%)	33 (17.6%)	97 (24.7%)
Local	C	30 (14.6%)	32 (17.1%)	62 (15.8%)
	D	41 (19.9%)	15 (8.0%)	56 (14.2%)
	E	25 (12.2%)	26 (13.9%)	51 (13.0%)
	F	15 (7.3%)	5 (2.7%)	20 (5.1%)
Total		206 (100%)	187 (100%)	393 (100%)

**Table 2 antibiotics-12-00613-t002:** Table showing the antimicrobial resistance profiles in recovered isolates.

Organism		Antibiotics													
	*N*	AMP	AMC	CAZ	CTX	FEP	FOX	CN	RD	NAL	CIP	SMX	TRIM	CHL	TET
*E. coli*	275	71 (26)	41 (15)	5 (2)	11 (4)	6 (2)	70 (25)	11 (4)	269 (98)	33 (12)	26 (9)	95 (35)	78 (28)	28 (10)	100 (36)
*Klebsiella* spp.	95	53 (58)	11 (12)	3 (3)	5 (5)	3 (3)	14 (15)	1 (1)	94 (99)	9 (9)	8 (8)	41 (43)	35 (37)	8 (8)	44 (46)
*P. aeruginosa*	2	2 (100)	2 (100)	0	2 (100)	0	2 (100)	0	2 (100)	1 (50)	0	2 (100)	2 (100)	1 (50)	2 (100)
*Salmonella* spp.	101	18 (18)	7 (7)	1 (1)	4 (4)	3 (3)	15 (15)	0	98 (97)	10 (10)	5 (5)	20 (20)	24 (24)	6 (6)	31 (31)
*Shigella* spp.	36	7 (20)	4 (11)	1 (3)	4 (11)	1 (3)	6 (17)	0	34 (94)	3 (8)	2 (6)	10 (28)	10 (28)	2 (6)	11 (31)
*Staph* spp.	42	37 (88)	21 (50)	21 (50)	21 (50)	21 (50)	20 (48)	3 (7)	32 (76)	13 (32)	0	15 (36)	15 (36)	6 (14)	24 (57)

**Table 3 antibiotics-12-00613-t003:** Table showing the isolation profiles based on antibiotics resistance class. The 14 antibiotics in this study were categorized into six classes based on CLSI 2021 guidelines. In addition, isolates resistant to three or more classes were considered MDR on the description of an earlier study by Basak et al. (2016) [28].

Organism		Number of Antibiotic Resistance Classes							Total
		0	1	2	3	4	5	6	
*E. coli*	Count	100	96	55	17	5	1	1	275
		(36.4%)	(34.9%)	(20.0%)	(6.2%)	(1.8%)	(0.4%)	(0.4%)	(49.9%)
*Klebsiella*	Count	21	39	20	8	4	2	1	95
		(22.1%)	(41.1%)	(21.1%)	(8.4%)	(4.2%)	(2.1%)	(1.1%)	(17.2%)
*Pseudomonas*	Count	0	0	0	1	1	0	0	2
		(0.0%)	(0.0%	(0.0%)	(50.0%)	(50.0%)	(0.0%)	(0.0%)	(0.4%)
*Salmonella*	Count	37	30	22	7	4	0	1	101
		(36.6%)	(29.7%)	(21.8%)	(6.9%)	(4.0%)	(0.0%)	(1.0%)	(18.3%)
*Shigella*	Count	15	11	6	2	2	0	0	36
		(41.7%)	(30.6%)	(16.7%)	(5.6%)	(5.6%)	(0.0%)	(0.0%)	(6.5%)
*Staphylococcus*	Count	4	2	4	3	9	18	2	42
		(9.5%)	(4.8%)	(9.5%)	(7.1%)	(21.4%)	(42.9%)	(4.8%)	(7.6%)
	Count	177	178	107	38	25	21	5	551
		(32.1%)	(32.3%)	(19.4%)	(6.9%)	(4.5%)	(3.8%)	(0.9%)	(100.0%)

## Data Availability

Data are available upon request from the corresponding author.

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
