# Peer review of "Antimicrobial Resistant Pathogens Detected in Raw Pork and Poultry Meat in Retailing Outlets in Kenya"

_antibiotics, 2023, doi:10.3390/antibiotics12030613_

Round 1
Reviewer 1 Report
The authors present data on Antimicrobial resistance pathogens detected in pork and poultry meat in retailing outlets in Kenya, the data presented is interesting but needs significant changes and revision. See my comments below.
Introduction:
Line 53: Can the authors right the full names before abbreviation i.e “LMICs”, this was the first introduction.
Line 89: “For instance” is repeated.
Line 117: Can the authors check this and either remove or re-word, its repetitive.
Results:
Section 2.1:
The authors indicate that the samples were acquired from various regions in Kenya, however, in the materials and results only five town were mentioned (Nairobi, Kisumu, Nakuru, Nanyuki, and Eldoret), there are 47 counties in Kenya and the samples were collected from roughly 5 counties, I believe this would give a skewed result. The authors also mention 75% of pork and 63% of poultry samples were collected from Nairobi, the study does not capture the full AMR profile of Kenya.
Section 2.3:
Can the authors include how the resistance percentage was calculated? Is the resistance percentage relative?
Can the authors also comment on why only 551 out of 661 samples were tested.
Discussion:
The authors should discuss that the majority of the samples were taken from fridge/freezer of fresh meat section suggesting cross contamination.
Line 281: The authors discuss that “…the present study showed exposure to antibiotics over time and virulence acquisition via mobile elements” – I suggest this is removed because the authors did not test for this, and no experiment was performed to confirm these claims.
Materials and Methods:
Section 4.2:
Can the authors please include a full method on how the samples were processed
The authors mention their enrichment method could detect fastidious organisms, however some very important pathogens such Campylobacter species (leading cause of foodborne enteritis) is a microaerophilic which requires special culturing, can the authors mention whether this was tested.
Author Response
Dear Reviewer1,
Thank you for finding our data interesting. Please find attached our responses to the reviewer's comments.
Kind regards,
Patrick Muinde

Reviewer 2 Report
The data are of some interest but the paper would require a major overhaul of the methods and results sections to make it acceptable. It is currently impossible to assess the reliability of the methods as so little information is provided.
Methods
Line 303: state here whether the meat samples were cooked or un-cooked before sale.
The methods section requires a lot more information. It is stated that “1g of the sample was homogenized in 9mL of buffered peptone water and Rappaport Vassiliadis enrichment media and incubated at 37°C overnight. The cultures were then plated on MacConkey, Xylose lysine de-oxycholate (XLD), and Mannitol salt agar”.
It is very unclear whether both types of broth were inoculated and sub-cultured or whether they were used sequentially. RV broth is used for the selective isolation of Salmonella. It cannot be reliably used for the recovery of other species. It is stated that broths were carefully chosen to support the growth of fastidious bacteria (line 318). How does buffered peptone water support the growth of fastidious bacteria. The suppliers and product codes of the broths should be provided.
Line 323 states that: “PCR and classical biochemistry, were utilized to determine the species of the isolates”. This does not enable readers to assess and/or repeat your work (which is the function of a materials and methods section). How can PCR be used to identify all possible strains that might be recovered without using dozens of primers (unless sequencing was also used?). Document the PCR primers that were used and also the exact nature of the biochemical tests.
A similar lack of detail is presented with respect to susceptibility testing. Which disc diffusion method was used? CLSI? EUCAST? What published breakpoints were used to determine whether strains were susceptible or resistant? This is critical information. What control organisms (with known susceptibility) were used to control the test?
Results
Line 148 states that 96.6% of poultry grew bacteria but line 151 states that the bacterial contamination rate of poultry was 50.9%. Please explain the difference between these two numbers.
Line 149 states that there were a total of 611 isolates. Line 166 states 661.
In my view it is not appropriate to combine resistance data for several different species into a single figure. Figures 2 and 3 should be deleted as they distort data of potential interest. For example, all S. aureus are resistant to nalidixic acid whereas rifampicin is not clinically useful against any of the Gram-negatives listed (there are no CLSI breakpoints that would indicate that E. coli, for example, is susceptible to rifampicin). Only acquired resistance is of interest. Pseudomonas is intrinsically resistant to at least half of the agents listed in Figure 3. Antibiotic resistance data should only be presented for each species and the only antibiotics that should be included for each species are those for which there are published zone diameter breakpoints (otherwise how to you determine whether a strain is susceptible or not?).
Why does the results section not document how many (if any) of the Staphylococcus strains were S. aureus (a foodborne pathogen)?
Line 347: why mention penicillin among the major conclusions? penicillin wasn't tested.
Minor amendments suggested
Line 56: change ‘resistance’ to ‘resistant’.
Line 62: re-phrase to read: “Several antibiotics classes are used in both humans and animals”
Line 70 I suggest changing ‘majorly’ to a different word e.g. ‘mostly’, ‘mainly’ or ‘especially’.
Line 75: change ‘forecasted’ to ‘forecast’ or ‘predicted’
Line 79: you are referring to consumption – therefore change ‘and are projected’ to ‘and is projected’.
Line 89 delete: ‘for instance’ (used twice)
Line 123: change ‘resistance’ to ‘resistant’.
Author Response
Dear Reviewer2,
Thank you for finding our data interesting. Please find attached our responses to the reviewer's comments.
Kind regards,
Patrick Muinde

Round 2
Reviewer 1 Report
I appreciate the authors have made changes to the manuscript.
Reviewer 2 Report
No further comments.